# Ultrasound Probe Pressure on the Maternal Abdominal Wall and the Effect on Fetal Middle Cerebral Artery Doppler Indices

**DOI:** 10.3390/medicina55080410

**Published:** 2019-07-26

**Authors:** Andrei Mihai Măluțan, Delia Clinciu, Ștefan Claudiu Mirescu, Răzvan Ciortea, Marina Dudea-Simon, Dan Mihu

**Affiliations:** 12nd Department of Obstetrics and Gynecology, “Iuliu Hatieganu” University of Medicine and Pharmacy, 400012 Cluj-Napoca, Romania; 2“Dominic Stanca” Obstetrics and Gynecology Clinic, “Iuliu Hatieganu” University of Medicine and Pharmacy, 400012 Cluj-Napoca, Romania; 3Molecular Biology and Biotechnology Department, Biology and Geology Faculty, Babes-Bolyai University, 400012 Cluj-Napoca, Romania

**Keywords:** fetal distress, pregnancy, middle cerebral artery, ultrasound, probe pressure

## Abstract

*Background and Objectives:* Doppler ultrasound of umbilical and fetal vessels is useful for monitoring fetal well-being, fetal anemia, intrauterine growth retardation, and other perinatal outcomes. The adverse perinatal outcome and circulatory changes can be reflected in fetal Doppler studies. The aim of this study was to evaluate the effect of increased pressure exerted on the maternal abdominal wall during routine ultrasound on the middle cerebral artery (MCA), resistance index (RI), pulsatility index (PI), and peak systolic velocity (PSV). *Materials and Methods:* A prospective study was conducted, in which we included 40 pregnant women between 24 + 0 and 41 + 3 gestational weeks (GW), with singleton pregnancies, without any associated pathologies, undergoing routine US examination. We recorded the flow velocity waveforms in the MCA, and we measured the RI, PI, PSV, and the applied pressure on to the maternal abdominal wall—needed for a proper evaluation of MCA. We then repeated the same measurements at two different higher pressure levels, at the same time having a proper image of the targeted vessel. *Results:* We found significant differences for the PI and RI levels with an increase in abdominal pressure (median PI 1.46, 1.58, and 1.92, respectively; median RI 0.74, 0.78, and 0.85, respectively; *p* < 0.05), for both PI and RI. At the same time, we found no significant differences for PSV in the studied group in relationship with increase in abdominal pressure (median PSV 39.56, 40.10, and 39.70, respectively; *p* > 0.05). *Conclusions:* The applied abdominal pressure by the examiner’s hand, during routine US scan in pregnancy, can modify the MCA parameters of blood flow resistance (PI and RI) when measured by Doppler US, thus influencing the diagnostic accuracy in a series of pregnancy associated pathologies, such as chronic fetal distress (CFD) or intrauterine growth restriction (IUGR).

## 1. Introduction

Fetal monitoring using pulsed wave Doppler for umbilical artery, anterior middle cerebral artery (MCA), ductus venosus, and uterine arteries is a routine method in obstetrical practice. It is well established that adverse perinatal outcome and circulatory changes can be reflected in fetal Doppler studies [1]. Doppler ultrasonography (Doppler US) of umbilical and fetal vessels is useful for monitoring fetal well-being, fetal anemia, intrauterine growth retardation (IUGR), and other perinatal outcomes [1]. Fetal anemia, caused mostly by maternal red cell alloimmunization, is evaluated by MCA peak systolic velocity (PSV) Doppler, which remains the gold standard for noninvasive screening of fetal anemia [2]. Alteration of MCA flow has also been associated with IUGR, intracranial anomalies, maternal-fetal hemorrhage, and twin-to-twin transfusion [3]. 

The cerebro-placental ratio (CPR) in fetal growth restriction is integrated in clinical practice and international guidelines [4,5]. A metaregression was performed to statistically compare the accuracy of the CPR, MCA Doppler, and umbilical artery (UA) Doppler. The results showed a lower sensitivity for MCA Doppler in comparison to UA Doppler and CPR in detection of fetal growth restriction. The CPR seems superior to UA Doppler in assessing fetal well-being in the case of fetal growth restriction [6].

In abnormal placental function, “brain-sparing effect” is the initial adaptive phenomenon. There is a cerebral vasodilation with vasoconstriction in the peripheral fetal vessels. The MCA pulsatility index (PI) and resistance index (RI) normal values in pregnancy have shown a parabolic curve, with a plateau between 28 and 30 weeks, likely due to increased requirement of blood supply to the brain during early and late pregnancy [1].

In the last trimester of pregnancy, a vasodilatation phenomenon appears in the fetal brain vessels. This physiological process decreases MCA PI values, which has been shown to precede the onset of spontaneous labor [7].

Many factors can influence MCA Doppler ultrasonography such as fetal behavior or fetal body movements and fetal breathing movements (FBM), uterine contractions, and measurement techniques [8]. A study performed on adults by Homburg et al. has noted that in case of increased intracranial pressure, the MCA Doppler parameters are modified [9]. At the same time, it was shown that the effect of maternal Valsalva maneuver did not alter the parameter values [10].

In the present study, we have investigated the relationship between pressure applied on the maternal abdominal wall during US examination and the MCA RI, PI, and PSV values, trying to assess if the exerted abdominal pressure would modify the abovementioned parameters, thus having a possible impact on the subsequent management.

## 2. Materials and Methods

### 2.1. Design

We conducted a prospective study in “Dominic Stanca” Obstetrics and Gynecology Clinic, Cluj-Napoca, Romania, in which we included 40 pregnant women between 24 + 0 and 41 + 3 gestational weeks (GW), with singleton pregnancies, without any associated pathologies, undergoing routine US examination. We excluded patients with any known pathologies that could affect fetal MCA Doppler parameters (preeclampsia, IUGR, fetal distress, ruptured membranes). The study design was approved by the Cluj Emergency County Hospital and “Dominic Stanca” Obstetrics and Gynecology Clinic, Cluj-Napoca Ethics Committee, with the decision 306/23.11.2017, and signed informed consent was received from each pregnant woman before US examination. The study was conducted under the tenets of the Helsinki Declaration.

The ultrasound machine used for patient evaluation was a Toshiba Aplio 300 (Toshiba Medical Systems Corp., Otawara, Japan), using a convex abdominal transducer (2–9 MHz). To minimize the intraoperator variability, all US examinations were conducted by a single operator (A.M.M.)—an obstetrics and gynecology specialist with an obstetric US subspecialty.

We examined the patients by transabdominal US, evaluating fetal lie and presentation, amniotic fluid index (AFI), area of placental insertion, and the abdominal wall thickness (AWT) including skin, subcutaneous tissue, and muscular wall. As a second step, we have evaluated the fetal cerebral vascularization, referring to the circle of Willis, with the help of color Doppler US, demonstrating the fetal MCA and recording the distance between the probe and the targeted vessel (e.g., MCA). By using pulsed Doppler examination, we recorded the flow velocity waveforms in the MCA, and measured the RI, PI, and PSV (using the highest point of the waveform). At the same time, we recorded, in a blinded manner, the applied pressure on to the maternal abdominal wall, considering this as a baseline pressure—needed for a proper evaluation of MCA. We then repeated the same measurements at two different higher pressure levels, while at the same time having a proper image of the targeted vessel. For the study methodology, we noted these pressure levels as level 2 and level 3. 

To measure the applied pressure, we performed an extrapolation between the applied abdominal pressure and the pressure exerted on the transducer by the examiner, using an electronic pressure sensor (FSR Interlink 402 round force sensor—Appendix A), which was attached directly onto the US probe, between the operators’ thumb and the transducer, without any direct contact with the maternal abdomen. The exerted pressure was recorded in units which can be converted in grams/mm^2^ (Appendix B).

Alpert et al. standardized certain parameters, like correct anatomical plane for image acquisition, the proper section of vessel for flow sampling, the proper angle, and the use of manual caliper placement versus auto-trace methods for obtaining the wave flow data [11]. In order to minimize the differences, we used these standard guidelines to perform Doppler US.

### 2.2. Statistical Analysis

The statistical analysis was performed using R 3.3.0. The normality of the data sets was checked using the Shapiro–Wilk test, the difference between samples was assessed using ANOVA test for the data following normal distribution and Kruskal–Wallis for the data not following the normal distribution. The statistical significance level was set to *p* < 0.05. 

## 3. Results

Table 1 and Table 2 present the descriptive statistics, demographic characteristics, and the abdominal pressure variables when studying MCA Doppler parameters. We recruited 40 patients, with a gestational age range between 24 + 0 and 41 + 3 weeks, having a median of 33 + 6 GW. As recorded, the mean AFI was 14 (range from 6.22 to 24.46, median 13.9) and mean BMI was 27.02 (range from 18.61 to 42.91, median 26.51). At the same time, the mean AWT was 12.64 mm, and the mean distance between the probe and the targeted vessel was 53.80 mm.

Regarding abdominal pressure variables, statistical analysis showed that the studied variables follow a normal distribution and there are significant differences between groups (ANOVA test, *p* < 0.05).

Figure 1, Figure 2, Figure 3, Figure 4, Figure 5, Figure 6 and Figure 7 and Table 3 show the differences between MCA RI, PI, and PSV measured in the studied group, applying three different levels of abdominal pressure. As shown in the subsequent tables and figures, there are significant differences for the PI and RI levels with increase in abdominal pressure (median PI 1.46, 1.58, and 1.92, respectively; median RI 0.74, 0.78, and 0.85, respectively). Data are not following the normal distribution, thus, we used a Kruskal–Wallis test, which confirmed *p* < 0.05, for both PI and RI. At the same time, we found no significant differences for PSV in the studied group in relation to the increase in abdominal pressure (median PSV 39.56, 40.10, and 39.70 respectively). Data are not following the normal distribution, thus, we used a Kruskal–Wallis test, which showed *p* > 0.05.

## 4. Discussion

In the present study, we have found a series of hemodynamic changes in the fetal brain, in relation to the exerted abdominal pressure during routine scan. We showed that the main markers used in monitoring the fetal brain hemodynamics, RI and PI, are significantly modified with the pressure applied on the abdominal wall by the examiner’s hand, observing a direct relationship between abdominal wall pressure and RI and PI values—and subsequently, a possible intrauterine fetal distress.

In order to explain the changes in Doppler indices with the increase in transabdominal pressure, several mechanisms should be considered. The pressure exerted by the transducer on the maternal abdomen is conducted through the abdominal wall, reaches the uterine wall and amniotic sac, spreads through the amniotic fluid, and is perceived by the fetus—this could generate a vagal reflex, manifested by bradycardia, which could be one of the underlying mechanisms responsible for the increase of MCA flow parameters [11]. On the other hand, in our study, we did not observe any significant change of the fetal heart rate in relationship with the pressure exerted on the maternal abdominal wall.

Another possible explanation would be to assume the increase of MCA PI and RI as a consequence of increased fetal intracranial pressure when applying external pressure with the transducer. In case of increased intracranial pressure, vessel caliber is reduced, leading to high peripheral vascular resistance and explaining the increase in Doppler indices in the MCA [10,12]. 

Abdominal wall thickness plays an important role in fetal MCA Doppler assessment, due to the greater or smaller distance that the ultrasound waves have to cross in order to reach the target. In case of a larger thickness of the abdominal wall, part of the applied external pressure will be damped in the wall, and part of the ultrasound will be absorbed before reaching the fetal brain [3].

Doppler examination of the fetal MCA is a widely used clinical parameter in assessing fetal well-being; however, few studies have been conducted so far, regarding the possible influence of the pressure applied with the transducer on the maternal abdominal wall on the variation of Doppler indices. Ramon et al. [13] applied external pressure on the abdominal wall, interrupting umbilical vein blood flow—as a consequence, they noted a decreased flow in the proximal MCA, with preservation of the flow in the other segments of the MCA and in the umbilical artery. Su et al. [10] evaluated the effects of both external abdominal pressure and maternal Valsalva maneuver on the fetal MCA RI, PI, PSV, and end diastolic velocity (EDV), showing a significant increase of the RI, PI, and PSV and decrease of the EDV following external pressure and no significant change in correlation with Valsalva maneuver. However, in both of these studies the pressure exerted on the maternal abdomen with the transducer was not quantified. Vyas et al. [14] used an open-ended water-filled manometer in order to measure the pressure applied with the transducer; they concluded that MCA PI significantly increased and the mean blood velocity significantly decreased with the increase of the amount of pressure. Finally, Alpert et al. [11] assessed the effect of transabdominal pressure on the fetal MCA-PSV using an electronic pressure sensor attached directly to the probe and found a positive correlation between the MCA-PSV and the level of pressure applied.

Our results regarding the increase of the MCA PI and RI with the exertion of higher transabdominal pressure are in accordance with the ones obtained by Su and Vyas [7,11]—however, as opposed to the results obtained by Su [10] and Alpert [11] revealing an increase in the MCA PSV, there was no significant difference noted in our study. The difference in group selection and US technique could be a possible explanation for the contradictory results. Our study has included patients with various presentation (cephalic, breech, or transverse), and this could possibly influence the results. Future studies may focus on each type of presentation, thus eliminating this variable.

To the best of our knowledge, this is the first study assessing the possible influence of the pressure applied with the transducer on the maternal abdominal wall on the variation of fetal MCA PI, RI, and PSV that uses a technique of pressure quantification.

At the same time, some limitations of this study must be addressed. In order to avoid maternal discomfort generated by a prolonged ultrasound examination, intra- and interobserver studies were not conducted. Another limitation is the relatively small number of subjects included in the study. Since there was a significant increase in the MCA PI and RI with the increase of transabdominal pressure applied, but not significant changes in the MCA PSV values, as opposed to the results obtained by Su [10] and Alpert [11], larger studies using the pressure quantification technique on these indices would be recommended. At the same time, it is well established that for best accuracy in fetal PSV assessment an insonation angle as close to 0 degrees as possible is needed. We did not achieve this parameter in all patients, and this could also limit the validity of our results. Also, the role of the abdominal wall thickness is a parameter that was not properly assessed in the present study, in relation to the measured indices, since the subjects in whom image acquisition was not optimal due to excessive abdominal wall thickness were not included in the study.

From a practical point of view, in the future, guidelines regarding a recommended maternal abdominal pressure could be elaborated, US probes could integrate pressure sensors, providing the possibility to monitor the pressure exerted on the maternal abdominal wall, and thus the examiner could adapt the US examination accordingly.

## 5. Conclusions

In conclusion, our study demonstrates a significant increase in the main Doppler markers used for monitoring the fetal brain hemodynamics, PI and RI, in relation to the examiners’ hand applied abdominal pressure. At the same time, we found no significant differences for PSV in relation to the increase in external abdominal pressure. These findings recommend caution when evaluating a possible brain sparing response to hypoxia, as an increased external abdominal wall pressure by the examiners’ hand could lead to a false negative result, and thus a delay in the proper management. At the same time, the results of the current study should be interpreted carefully, as larger studies are needed for a better understanding on how the fetal hemodynamics are iatrogenically influenced.

## Figures and Tables

**Figure 1 medicina-55-00410-f001:**
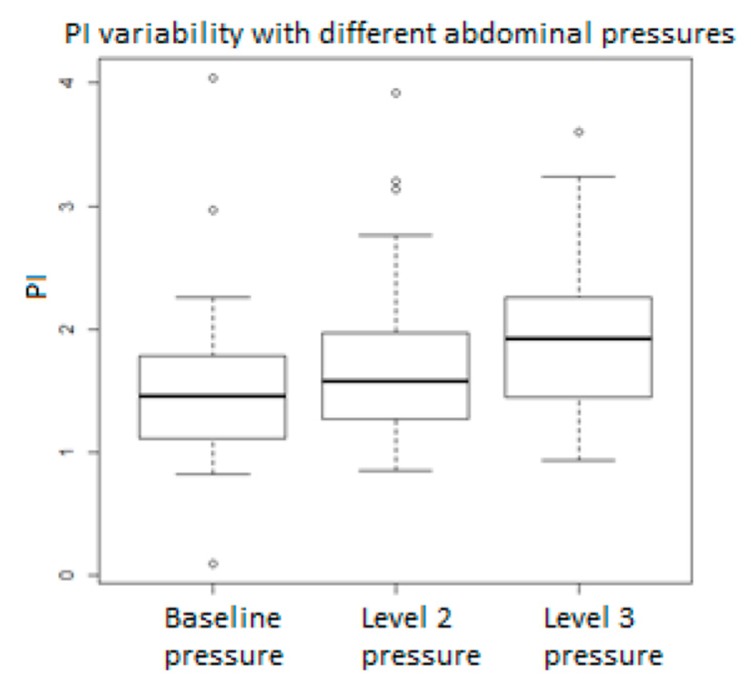
Pulsatility index (PI) values in relation to abdominal pressure.

**Figure 2 medicina-55-00410-f002:**
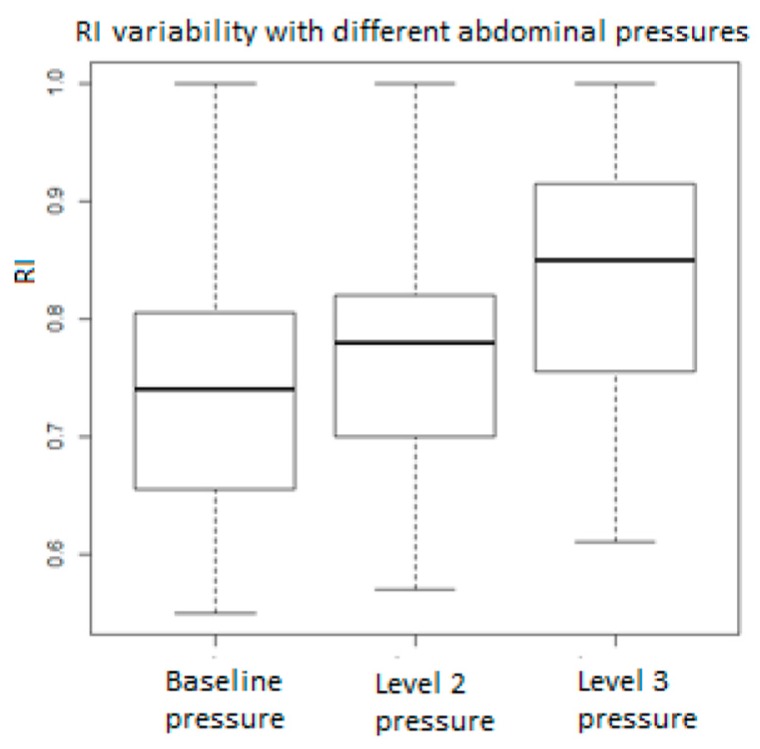
Resistance index (RI) values in relation to abdominal pressure.

**Figure 3 medicina-55-00410-f003:**
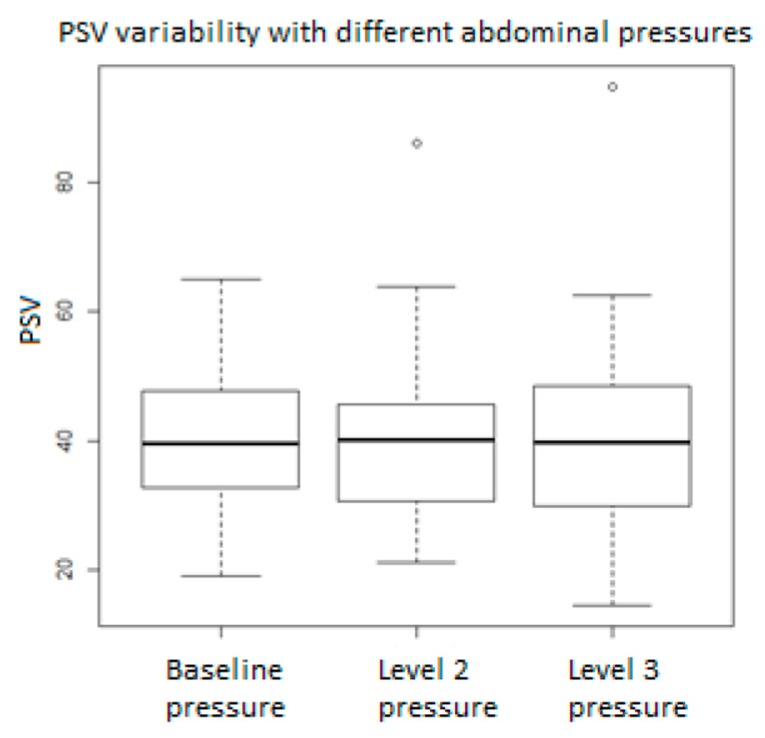
Peak systolic velocity (PSV) values in relation to abdominal pressure.

**Figure 4 medicina-55-00410-f004:**
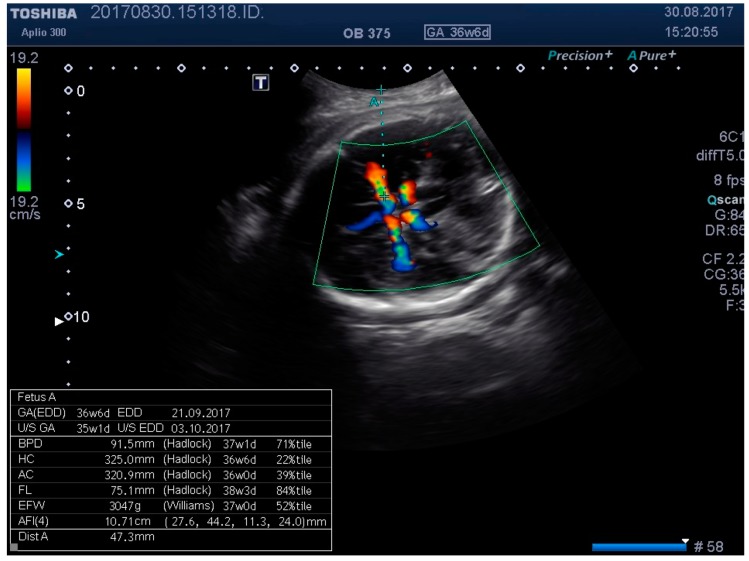
Middle cerebral artery (MCA) identification and measuring the distance to vessel.

**Figure 5 medicina-55-00410-f005:**
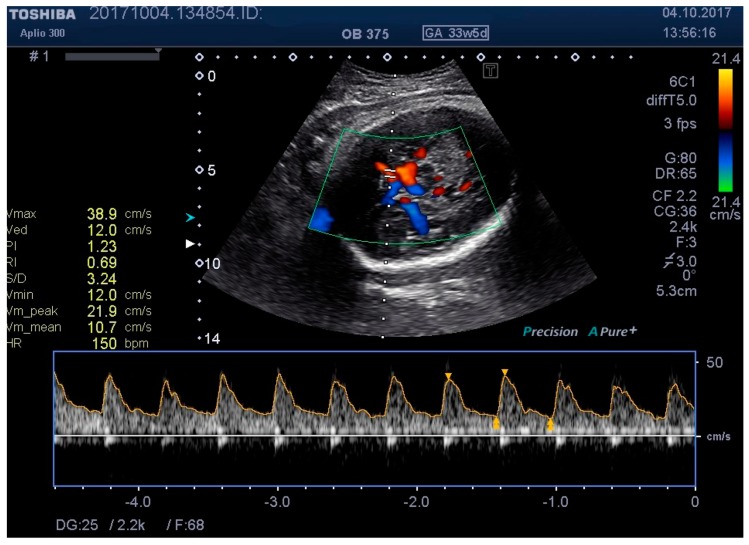
MCA Doppler spectrum with baseline pressure.

**Figure 6 medicina-55-00410-f006:**
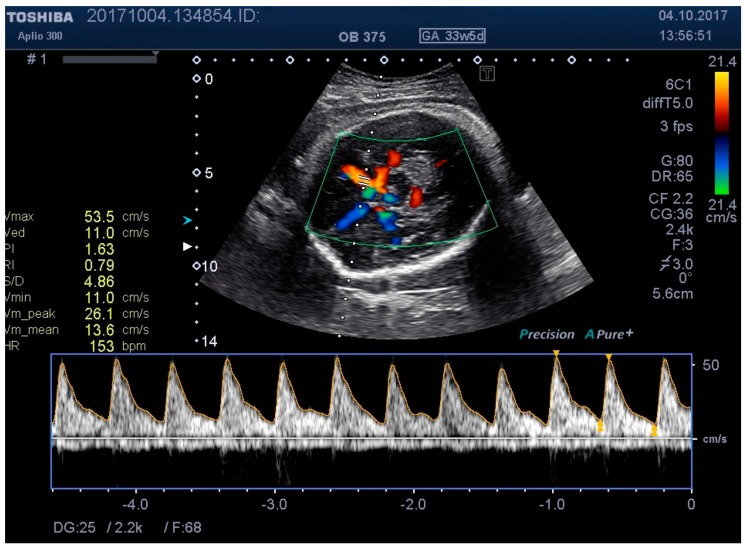
MCA Doppler spectrum with level 2 pressure.

**Figure 7 medicina-55-00410-f007:**
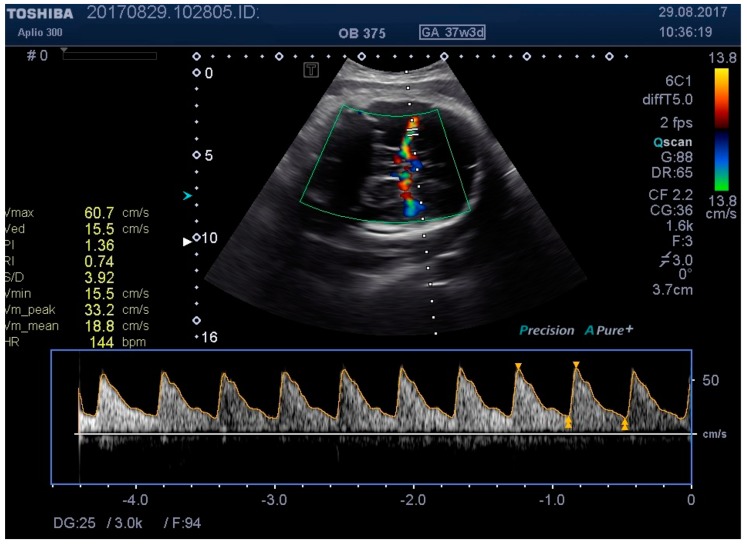
MCA Doppler spectrum with level 3 pressure.

**Table 1 medicina-55-00410-t001:** Descriptive statistics and demographic characteristics of the studied group.

Studied Parameter	Mean	Standard Deviation	Median	Min	Max	Number
Gestational age (weeks)	33 + 2	5 + 3	33 + 6	24 + 0	41 + 3	
AFI	14	3.93	13.90	6.22	24.46	
BMI	27.02	4.87	26.51	18.61	42.91	
AWT (mm)	12.64	5.86	10.65	4.60	25.70	
Distance to vessel (mm)	53.08	9.09	51.65	38.40	79.00	
Fetal presentation						
Vertex						34
Breach						2
Transverse						4
Parity						
Nullipara						24
Multipara						16
Placenta						
Uterine fundus						27
Anterior						7
Posterior						6

AFI—amniotic fluid index; BMI—body mass index; AWT—abdominal wall thickness.

**Table 2 medicina-55-00410-t002:** Abdominal pressure variables for the studied group.

Parameter	Baseline Pressure	Level 2 Pressure	Level 3 Pressure
Mean	105.78	206.45	299.88
Standard Deviation	32.48	43.61	40.27
Min	50	97	200
Max	185	270	389
Range	135	173	189

**Table 3 medicina-55-00410-t003:** PI, RI, and PSV variability in the studied group in relation to abdominal pressure.

**Pulsatility index**	**Baseline Pressure**	**Level 2 Pressure**	**Level 3 Pressure**	***p*-Value**
Median	1.455	1.580	1.920	0.005492
Min	0.10	0.85	0.94	
25% Quart	1.13	1.28	1.47	
75% Quart	1.77	1.94	2.24	
Max	4.04	3.92	3.60	
IQR	0.65	0.67	0.77	
**Resistivity index**	**Baseline Pressure**	**Level 2 Pressure**	**Level 3 Pressure**	***p*-Value**
Median	0.740	0.780	0.850	0.002072
Min	0.55	0.57	0.61	
25% Quart	0.66	0.70	0.76	
75% Quart	0.80	0.82	0.91	
Max	1	1	1	
IQR	0.15	0.12	0.15	
**Peak systolic velocity**	**Baseline Pressure**	**Level 2 Pressure**	**Level 3 Pressure**	***p*-Value**
Median	39.55	40.10	39.70	0.9595
Min	19.18	21.10	14.61	
25% Quart	32.82	30.86	30.15	
75% Quart	47.15	45.55	47.50	
Max	64.9	86.0	94.7	
IQR	14.30	14.69	17.36

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
