# Peer review of "Ultrasound Probe Pressure on the Maternal Abdominal Wall and the Effect on Fetal Middle Cerebral Artery Doppler Indices"

_1010-660X, 2019, doi:10.3390/medicina55080410_

Round 1

Reviewer 1 Report

I would like to congratulate to all the authors regarding well written manuscript .

Introduction: 

I recommend to shortening the Introduction. This is not a teaching book. 

66 - 73 I suggest to move to  Methods with some changes.

44 References ? All sentences should be supplied with the References

Materials

80 feta -fetal

Discussion

160 this could generate a vagal reflex, manifested by bradycardia, Could you provide any data from your study regarding fetal bradycardia under probe pressure?

Have you any idea how could be possible to implement this ,, pressure management, checking'' in the practical work? If yes, it would be nice if you could describe this or just to discuss a bit more about it . 

Author Response

Response to Reviewer 1.

Thank you for reviewing our manuscript and for all your comments; they were very constructive.

As a response:

·        The manuscript has been revised regarding English language and style

·        Introduction has been shortened

·        The paragraph mentioned (66-73) has been partially moved in Methods and rewritten. We kept at the end of Introduction, the scope of the study.

·        All sentences or paragraphs have References, we have also introduced a new reference (e.g. 2) and renumbered all the others.

·        Have corrected the term feta to fetal

·        We have introduced a statement regarding fetal bradycardia in our study, explaining that we did not observe any significant changes

·        We have also introduced a statement at the end of Discussion regarding a possible practical role in the future, for abdominal wall pressure monitoring.

All changes are highlighted in yellow, and with strikethrough.

Thank you again and best regards,

Authors

Reviewer 2 Report

Abstract - line 19, I believe RI is the abbreviation for "Resistance index" rather than "Resistivity" (and line 55)

abstract conclusion - rather than saying that the pressure modifies MCA Doppler parameters, could the authors be more specific and state that pressure modifies the MCA parameters of blood flow resistance (which incorporates PI and RI) when measured by Doppler, since PSV is not affected - likewise, diagnostic accuracy fetal anaemia shouldn't be listed as affected by abdominal pressure given that PSV is not affected

Reference is needed for lines 48-49 - which international guideline have integrated CPR?

Lines 49-52: The statements regarding sensitivity of MCA Doppler compared to CPR and UA Doppler are simplistic, as the benefits of one over another are gestation specific and this should be discussed with more clarity

Line 66 - "were" should be used instead of "was" given that "guidelines" are plural

Line 67 - "plain" should instead be "plane"

Line 80 - the L is missing from the word "fetal" - instead written as "feta"

line 102 - examiners should not have an apostrophe between the r and s

Is the abbreviation AWT defined in the text? I only found the definition in the figure of a table

Could the p values be included in the table? Were the Doppler parameter levels compared between level 1 and level 2, level 1 and level 3, and level 2 and level 3, or altogether from 1, 2 and 3 progressively?

From the images it appears that no angle correction was made for the MCA vessel to achieve closer to 0 degrees? Is this correct? For PSV assessment especially, angle of insonation needs to be zero degrees

"examiners" in line 154 should be "examiner's"

The authors mention fetal bradycardia in response to abdominal wall pressure as a potential mechanism, but did they notice a decrease in fetal heart rate during their assessments? Was this analysed?

The authors note that the fact that PSV values were not altered by pressure is in contrast to previous studies - please expand on why this might be the case

Author Response

Response to Reviewer 2.

Thank you for reviewing our manuscript and for all your comments; they were very constructive.

As a response:

·        The manuscript has been revised regarding English language and style

·        Resistivity index has been changed to resistance index throughout the text

·        Abstract conclusion has been rewritten accordingly

·        We have introduced 2 new references to the CPR paragraph in Introduction. Also, we have reformulated the paragraph, to be clearer. We did not discuss the differences between CPR, PI, and RI sensitivity more widely, as we consider it was not appropriate for the current article, and in regardence with the recommendation to keep the length of the introduction. All info, is present in the article provided as Reference.

·        Line 66, 67, 80 have been corrected or reformulated.

·        We have defined AWT in text

·        P-values have been included in table 3

·        We have also included a statement in study limitations regarding angle of insonation

·        Examiners has been changed to examiner’s

·        We have introduced a statement regarding fetal bradycardia in our study, explaining that we did not observe any significant changes

·        We introduced a sentence trying to explain the contradiction with the results of other studies

All changes are highlighted in yellow, and with strikethrough.

Thank you again and best regards,

Authors
